# Relation between Photogrammetry and Spinal Mouse for Sagittal Imbalance Assessment in Adolescents with Thoracic Kyphosis

**DOI:** 10.3390/jfmk8020068

**Published:** 2023-05-19

**Authors:** Guido Belli, Stefania Toselli, Mario Mauro, Pasqualino Maietta Latessa, Luca Russo

**Affiliations:** 1Department of Sciences for Life Quality Studies, University of Bologna, 47921 Rimini, Italy; guido.belli@unibo.it (G.B.);; 2Department of Human Sciences, IUL Telematic University, 50122 Florence, Italy; l.russo@iuline.it

**Keywords:** kyphosis, spinal mouse, photogrammetry, kinesiology, postural evaluation

## Abstract

The evaluation of postural alignment in childhood and adolescence is fundamental for sports, health, and daily life activities. Spinal Mouse (SM) and photogrammetry (PG) are two of the most debated tools in postural evaluation because choosing the proper instrument is also important to avoid false or misleading data. This research aims to find out the best linear regression models that could relate the analytic kyphosis measurements of the SM with one or more PG parameters of body posture in adolescents with kyphotic posture. Thirty-four adolescents with structural and non-structural kyphosis were analyzed (13.1 ± 1.8 years; 1.59 ± 0.13 m; 47.0 ± 12.2 kg) using SM and PG on the sagittal plane in a standing and forward-bending position, allowing us to measure body vertical inclination, trunk flexion, and sacral inclination and hip position during bending. The stepwise backward procedure was assessed to estimate the variability of the grade of inclination of the spine and thoracic spine curvature with fixed upper and lower limits, evaluated with SM during flexion. In both models, the PG angle between the horizontal line and a line connecting the sacral endplate–C7 spinous process and the PG hip position were the best regressors (*adjusted-R*^2^ SM bend = 0.804, *p* < 0.001; *adjusted-R*^2^ SM fixed bending = 0.488, *p* < 0.001). Several Spinal Mouse and photogrammetry parameters showed significant correlations, especially when the Spinal Mouse measurements were taken when the adolescents were in the forward-bending position. Physicians and kinesiologists may consider photogrammetry as a good method for spinal curve prediction.

## 1. Introduction

Postural alignment in childhood and adolescence can be considered one of the most important sources of worry for parents, with particular attention being paid to the spine. Altered postural alignments can be classified as structural and non-structural misalignments, even if the postural appearance of these disturbances may be similar [1]. Structural misalignments indicate the presence of morphological abnormalities within the bones and soft tissues. Conversely, non-structural misalignments do not show any bone disorder but evidence a non-anatomic spine alignment with a moderate-to-good degree of self-correction [2]. Both structural and non-structural misalignments can affect the sagittal balance of the spine [3,4,5].

With specific regard to the sagittal plane, several classifications have been previously reported [2,6,7,8,9]. Within these classifications, kyphotic posture is frequently and easily recognized by parents. The thoracic spine in children and adolescents can be defined in a normal range as being between 20–40° degrees [1], and hyperkyphosis diagnosis is considered beyond 45° [10,11]. Nevertheless, other authors suggest that the average values of thoracic kyphosis are 42.0° ± 10.6° and 45.8° ± 10.4° from a cohort of 167 children of 8.1 ± 2.0 years and 479 adolescents of 13.6 ± 1.9 years, respectively [12].

Measuring procedures in posture can be widely utilized, such as with posturography, to analyze the effect of specific interferences on postural control [13,14,15]; or they can be analytical and descriptive, such as with photogrammetric analysis of the spine, with a focus on sagittal balance assessment using non-radiographic methods as an example [16]. In the last 15 years, the main three non-radiographic methods for spine evaluation were rasterstereography, skin-surface mouse, and photogrammetry [17,18,19,20]. Rasterstereography is mainly represented by two kinds of measuring methods: (1) the first one uses the analysis of the light projected on the subject’s skin; it is reliable and represents the most widespread solution for the application of rasterstereography [21,22,23]; (2) the second one uses an infrared and time-of-flight 3D RGB camera, and it seems to be reliable as well [24]. The skin-surface mouse is mainly represented by Spinal Mouse^®^ (IDIAG, Fehraltorf, Switzerland), a valid and reliable tool for spine assessment, in particular for kyphotic posture [25,26], that can be rolled along the profile of the spine measuring the vertebral shape and angulation [16,27,28,29]. Finally, photogrammetry, in particular 2D modality, is one of the most used and cheapest tools for kinematic and geometrical analysis of motion and even posture. It can be performed using different software, and many of them are valid and reliable [30,31,32]. All these non-radiographic methods evidence some advantages but show several limits for spine evaluation. Researchers, but even more so professionals, should always balance the cost/effectiveness ratio as well as the ease and accuracy of the measurement. Rasterstereography usually offers a wide range of postural parameters; it is very fast to use but represents a high-cost tool. Spinal Mouse has a lower cost, but the price range is not accessible for all; the accuracy is high, as is the software analysis, but this is only focused on the spine [25,27]. Finally, photogrammetry is the cheaper one, as well as the less “smart”. Depending on software features and user skills, photogrammetry can be more or less “user friendly”; at the same time, it also allows the user to obtain measures of the whole body, not only of the spine [32].

Within this scenario, it could be very interesting for professionals to use photogrammetry, the easiest and cheaper way, as a global screening tool to detect in advance the signs of a spine misalignment, with particular regard to the kyphosis curvature. Considering the whole body as a kinetic chain [33], and that photogrammetry could easily measure the alignment of the whole body with good validity and reliability [32], a relation could be hypothesized between thoracic spine behavior and one or more photogrammetric measurements of body posture. Although radiographic and photogrammetric procedures have been previously investigated in order to quantify thoracic kyphosis and lumbar lordosis [34,35], the comparison between Spinal Mouse^®^ evaluation and photogrammetry is lacking. Since Spinal Mouse^®^ can be used in several body positions (upright standing, forward trunk bending, seated side-bending, for example), additional information could be obtained from this device. Therefore, this study aims to find out a model of regression that could relate global and analytic measurements of the Spinal Mouse^®^ on the sagittal plane with one or more photogrammetric parameters of body posture in adolescents with kyphotic posture.

## 2. Materials and Methods

### 2.1. Design and Participants

This is a cross-sectional study design. Participants were recruited from Fisiokinè Medical Centre (Scandiano, Reggio Emilia, Italy). The criteria of selection included a diagnosis of increased thoracic kyphosis (postural or structural hyperkyphosis), no history of musculoskeletal or neurological pain in the last 3 months, no prior surgical intervention for spine disorders, and aged between 10 to 16 years old. No gender restrictions were defined. All participants were informed and gave voluntary consent to participate in the study. Parents’ consent was requested, since participants were younger than 18 years old. The privacy criteria were met. The study was approved by the Bioethics Committee of the University of Bologna and was conducted in accordance with the guidelines of the Declaration of Helsinki; the project identification code was n.2.18.

During the recruitment phase, each participant completed the anamnesis investigation. All specific medical reports were collected and analyzed to meet the selection criteria. The enrolment phase lasted 6 months, from June to December 2022.

### 2.2. Measurements Instruments

#### 2.2.1. Spine Analysis

To evaluate spinal curves and trunk alignment, the SpinalMouse^®^ (IDIAG M360^®^, Mülistrasse 18, CH-8320 Fehraltorf, Switzerland) device was used. It is a non-invasive computer-assisted medical device that quantifies the curvature and mobility of the spinal column in the frontal and sagittal planes by gliding manually along the spine [28,36]. Data are sampled every 1.3 mm while the mouse is rolled from vertebra C7 to S3, giving a sampling frequency of approximately 150 Hz. Results are wirelessly transferred to a computer, where the IDIAG software displays vertebral positions, joint angles, and spinal measurements. A recent study reported a high correlation between Cobb angle evaluated with X-ray and intra (ICC = 0.872) or inter-observer (ICC = 0.962) SpinalMouse^®^ measurements on the frontal plane [37]. In addition, this device evidenced excellent intra-rater reliability for the analysis of sagittal thoracic and lumbar curvature and mobility in hyperkyphosis [25]. In the present study, the SpinalMouse^®^ measurements were performed by a trained specialist with more than five years of experience. Data were collected in a quiet and well-lit environment with a comfortable temperature [38,39]. The evaluation was settled in the morning to avoid positional differences in the spine due to fatigue and/or daily stress factors. After undressing the upper body, the C7–S3 vertebral spinal processes were determined and marked with a dermographic pen by the specialist while the patient was standing up in the anatomical position. Measurements were performed in 3 different trunk positions during standing: neutral, maximal flexion, and extension (sagittal plane evaluation). In the neutral position, the participant was asked to maintain a relaxed position, looking and facing horizontally toward the wall, with the feet shoulder-width apart and with straight knees and arms by the side. In maximal flexion, the subject was asked to flex the trunk with extended legs as far as possible, aiming to touch the ground with fingertips. In maximal extension, the participant was asked to cross their arms in front of the chest and extend the trunk as far as possible, without extension of the cervical spine. SpinalMouse^®^ was then moved downwards along the spinal criteria points, in each position. Participants did not perform a warm-up before the examination. Some specific measures were extracted and analyzed from all raw data available. The eight variables were: the inclination of the spine in standing (SM stand); the inclination of the spine during flexion (SM bend); thoracic spine curvature with fixed upper and lower limits (first and last thoracic vertebra) in standing (SM fixed stand) and during flexion (SM fixed bend); thoracic spine curvature with physiological upper and lower limits (defined by Spinal Mouse software) in standing (SM phys. stand) and during flexion (SM phys. bend); and spine length in standing position (SM Rachid stand) and during flexion (SM Rachid bend). Figure 1 shows some of all Spinal Mouse possible measurements displayed using IDIAG M360 software and used in the present study.

#### 2.2.2. Photogrammetric Postural Analysis

Postural evaluation using photogrammetry has been previously demonstrated to be a reliable method in young people with postural misalignments [40,41]. Recently, photogrammetric measurements of thoracic kyphosis showed excellent test–retest reliability (ICC = 0.97; SEM = 1.67; MDC = 4.62) in adolescents with hyperkyphosis and evidenced a strong correlation between the values obtained with this technique and radiography methods [42]. In the present study, 2 digital photographs (standing right-side and standing with trunk flexion) were recorded using a portable device (Tablet Huawei^®^ Mediapad, Huawei Base, Bantian, Longgang District, Shenzhen, China) to analyze the sagittal plane. The device was set on a tripod, three meters away from the line marking the position of the participant. The height of the tripod was adjusted so the middle of the objective lens was 100 cm above the ground [43]. Each participant was initially positioned in front of the camera with a postural grid (ATS^®^, Largo Cairoli 10, 52100 Arezzo, Italy) on the back, then made to turn their body to the left to show the right side perpendicular to the lens, with feet placed in a fixed position over a specific area (standing right-side position—Figure 2A). Successively, participants flexed the trunk and remained in the forward-bending position (standing trunk flexion position—Figure 2B–D). The APECS-AI Posture Evaluation and Correction System^®^ (New Body Technology SAS, 12 Rue Pierre Semard, Incubagem 38000 Grenoble, France) was used to evaluate absolute and relative angles in the sagittal plane [29,32]. Specifically, the following angles were investigated: body vertical inclination (absolute angle between the vertical line and a line connecting the lateral malleolus–tragus of the ear: PG mall–tragus); trunk flexion (absolute angle between the horizontal line and a line connecting sacral endplate—C7 spinous process, PG trunk bend); sacral inclination during bending (absolute angle between horizontal line and a tangent line to the sacral dorsum, PG sacrum bend); hip position during bending (absolute angle between the vertical line and a line connecting the lateral malleolus–greater trochanter, PG hip bend). To better detect previous anatomic landmarks during photographic analysis, an adhesive tape was applied to the skin [43].

Figure 2 shows the four angles analyzed with the APECS application.

### 2.3. Statistical Procedure

The descriptive statistics were reported as the mean, standard deviation (std), minimum (min), and maximum (max) values for each variable. The variables’ distribution was verified with the Shapiro–Wilk test. The Pearson product–moment (r) was calculated to measure the degree of correlation between the variables. To perform the best regression model, the stepwise backward procedure was assessed with a significant level for entry or removal to or from the model equal to 0.10. The model’s heteroskedasticity was checked using the Breusch–Pagan/Cook–Weisberg test. The multicollinearity was checked using the variance inflation factor (VIF), and a value lower than 5 was considered acceptable (moderate correlation) [44]. The Cook’s distance plot was performed to look for the outlier presence, with a threshold settled at n/4. If one or more outliers affected the model, they were removed, and a new model was performed. The adjusted R2 was calculated to report the goodness-of-fit for the proposed model. Additionally, the F value, the root MSE, the regression coefficient (β), the standard error, the student’s *t*-test value, and the 95% confidence interval were reported. The significance level was settled at ≤0.05. Finally, the Bland–Altman plot, the pairwise correlation, and the concordance correlation coefficient (CCC) were computed to compute the degree of agreement between the Spinal Mouse and photogrammetry [45].

## 3. Results

Table 1 shows the descriptive statistics for all the variables. No missing value was met for all variables (n = 34). Similar values were found between fixed and physiological Spinal Mouse evaluation, in both standing (45.94 ± 8.24 and 48.29 ± 9.14) and bending (63.71 ± 10.23 and 63.41 ± 9.45) positions. The highest grade was found in the inclination of the spine during flexion (122°), whereas the smallest value was the grade of the inclination of the spine in a standing position (−6°).

Table 2 shows the variables’ correlation matrix. Generally, high Pearson correlation coefficients between Spinal Mouse and photogrammetry measurements were found in the bending position (*p* < 0.05). In particular, a wide positive correlation appeared among the inclination of the spine during trunk flexion and trunk flexion, calculated as the absolute angle between the horizontal line and a line connecting the sacral endplate–C7 spinous process (r = 0.839, *p* < 0.001). Differently, the grade of the inclination of the spine during flexion decreased with the increasing of the absolute angle between a horizontal line and a tangent line to the sacral dorsum (r = −0.732, *p* < 0.001). Additionally, the thoracic spine curvature with fixed upper and lower limits in the bending position showed high correlations with the trunk flexion and the sacral inclination detected using photogrammetry (*p* < 0.001).

### Linear Regression Models

Table 3 shows the result of the stepwise procedure. Two outliers were removed from the first model, and the two photogrammetry measurements in bending, such as the angle between the horizontal line and a line connecting the sacral endplate–C7 spinous process and the hip position, explained 80.4% of the variability of the spine inclination during trunk flexion measured with the Spinal Mouse on 32 adolescents. The Breusch-Pagan/Cook-Weisberg test accepts the null hypothesis of heteroskedasticity absence (*χ*^2^_(1)_
*=* 0.77, *p* = 0.711). The mean VIF was 1.19.

Figure 3 shows the scatterplots with the errors of bending SM and each regressor, respectively.

Figure 4 shows the Bland–Altman graph (A) and scatterplot with Spinal Mouse and the new model (B). The concordance correlation coefficient was 0.995 and Pearson’s r = 0.904 (mean = 90.125 ± 12.067).

The new equation to estimate the Spinal Mouse degree in bending is
*SM bend* = (0.86413 ∙ *PG hip bend*) + (1.0557 ∙ *PG trunk bend*) − 12.749


Table 4 shows the result of the stepwise procedure on the thoracic spine curvature with fixed upper and lower limits, in the bending position. Three outliers were removed from the first model, and the two photogrammetry measurements in bending explained 48.79% of the Spinal Mouse fixed kyphosis variability in 31 adolescents. The Breusch–Pagan/Cook–Weisberg test accepts the null hypothesis of heteroskedasticity absence (*χ*^2^_(1)_
*=* 1.80, *p* = 0.179). The mean VIF was 1.32.

Figure 5 shows the scatterplots with the errors of SM fixed kyphosis measured in bending position and each regressor, respectively.

Figure 6 shows the Bland–Altman graph (A) and scatterplot with Spinal Mouse fixed kyphosis and the new model (B). The concordance correlation coefficient was 0.686 and Pearson’s *r* = 0.723.

The new equation to estimate the Spinal Mouse fixed kyphosis degree in the bending position is
*SM fixed bending =* (−2.122705 *∙ PG hip bend*) *+* (−0.678946 *∙ PG trunk bend*) *+* 144.2784


## 4. Discussion

The present study aimed to correlate photogrammetry and Spinal Mouse^®^ during the postural evaluation of adolescents with a diagnosis of structural or non-structural hyperkyphosis. Current findings evidence a positive correlation between some measurements of standing trunk flexion performed with both devices and highlight how photogrammetry could explain 80.4% of Spinal Mouse variability during forward bending. To the best of our knowledge, this is the first study that compared SM and PG.

In recent years, several authors analyzed the reliability and validity of non-radiographic methods during sagittal balance assessment in different populations. Since an adequate anterior–posterior balance condition is fundamental to maintaining an upright, efficient, and painless posture, the evaluation of the sagittal profile has gained much relevance in spinal pathology [46]. In this direction, Cohen et al. [16] reported that plumbline, surface topography, infrared motion analysis, and SM show moderate-to-high validity and reliability. In their systematic review of 14 articles, authors suggested that these methods can be a non-invasive approach to monitor global sagittal balance, even if specific limitations are present and spinopelvic parameters represent the “gold standard” (sacral vertical axis, pelvic tilt, and sacral slope, as an example). Furthermore, Barret et al. [26] evidenced that SM, the Debrunner kyphometer, and the Flexicurve index have the strongest levels of reliability among 15 non-radiographic methods to assess thoracic kyphosis. Starting from this consideration and previous research [25,47,48], SM was included in our study. Roghani et al. evaluated SM reliability on a sample of women with and without hyperkyphosis (aged between 60–80 years), with a focus on thoracic and lumbar curvature, pelvic position, trunk inclination, and spine mobility. The results evidenced a high intra-rater reliability for all measurements in both groups (ICC: 0.89–0.99), with standard error of the means (SEMs) ranging from 1.02° to 2.06° and from 1.15° to 2.22° in the hyperkyphosis and normal group, respectively. In addition, the minimal detectable change (MDC) ranged from 2.85° to 5.73° in the hyperkyphosis group and from 3.20° to 6.17° in the normal group. The authors concluded that SM is a useful, easy, and low-risk device to assess spinal curvature and mobility. Demir et al. evaluated SM test–retest reliability in 28 female adolescents (aged between 15–18 years) during upright standing on the frontal and sagittal planes. Their results evidenced good reliability for thoracic and lumbar curvature on the sagittal plane and confirmed the use of these tools for “in-field” screening and clinical assessment. Muyor and collaborators analyzed the Sit-and-Reach Test and Toe-Touch test (same as the forward bending test) using SM to define the criterion validity of both tests during hamstring flexibility assessment. The study involved 141 athletes from different sports (tennis players, kayakers, canoeists, cyclists), all aged between 15–17 years. Research findings suggested that pelvic tilt and lumbar motion have a greater impact on test scores than hamstring flexibility (measured with a passive straight leg test). In the present study, only some SM measurements were investigated. Specifically, fixed and physiologic thoracic curvature, body inclination, and spine length in upright and bending positions were chosen. The reason is mainly related to: (1) the sample features (adolescents with postural or structural hyperkyphosis); (2) to compare these parameters with some specific photogrammetric variables; (3) to attempt to find a few quick and easy-to-detect photogrammetric landmarks [31,49]. Mean values for SM thoracic curvature in a standing position were 45.9° and 48.2° for fixed and physiological kyphosis, respectively. These values are slightly above the reported range for normal curvature (20–40°) and evidence of slight hyperkyphosis. Similar values have been reported in different sports players performing in flexed positions (kayak, canoa, tennis) and that were aged between 15–17 [50], as well as for adolescents aged between 12–15 [51]. Anyway, it must be considered that higher values have been found for structural spine misalignment diagnosis (range of 50–62°) compared to non-structural (range of 38–50°).

For photogrammetric analysis, the application APECS-AI Posture Evaluation and Correction System^®^ has been used. This tool is an easy and low-cost program that allows one to assess body posture in different positions. Recently, Trovato et al. [32] evaluated a sample of 50 males and 50 females (mean age 23.4 years) to investigate gender differences in anterior coronal, posterior coronal, and sagittal planes. Their results evidenced good reproducibility for most of the 24 variables analyzed and reported some gender-related features. In the present study, only four sagittal parameters were investigated to assess the anterior–posterior balance in upright and bending positions. This selection has been defined for the abovementioned reasons. In particular, trunk inclination in both positions and pelvic motion during forward bending have been correlated with thoracic curvature. As regards body alignment, the results evidenced a mean value of 2.7° and 91.7° during standing or bending positions, respectively. Sagittal vertical inclination (upright posture) is similar to the reference value of 2.6° highlighted by Trovato et al. in their sample and superior to the reference value of 1.73° defined by Krawky et al. [31]. The difference could be related to the photogrammetry techniques reported in both studies [43,52]. Results did not show a correlation between PG and SM in relation to this parameter (2.6° vs. 1.6°, respectively). Since the body vertical alignment using PG was calculated by using the absolute angle between the vertical line and a line connecting the lateral malleolus–tragus of the ear, while SM evaluated it from the C7–S3 connection and vertical line, this result could be expected. Conversely, a significant positive correlation was found between the two devices during the bending position. In this postural assessment, PG and SM used a similar technique (absolute angle between the horizontal line and a line connecting the sacral endplate–C7 spinous process for PG or absolute angle between the vertical line and a line connecting the S3–C7 spinous process for SM). The landmarks for pelvic parameters during forward bending were defined as reported by Carregaro et al. [53]. Since adolescents with hyperkyphosis show a low level of flexibility across the muscular posterior chain, sacral inclination and hip position were calculated to investigate pelvic displacement in the sagittal plane [6,33,48]. PG Sacrum inclination evidenced a significant correlation with SM trunk inclination during bending, and linear regression reported that trunk bending added to hip bending explained 80.4% of SM variability in relation to this parameter (first model). Furthermore, the same PG variables explained 48.7% of SM variability concerning thoracic curvature during bending. These results highlight how the photogrammetric analysis of the pelvic region is deeply connected with spine inclination in this kind of subject.

Since the present study aims to correlate quick and easy-to-detect PG measurements with Spinal Mouse^®^, the abovementioned variable was chosen. Previous researchers analyzed different pelvic parameters using photogrammetry (pelvic horizontal alignment and hip angle, for example), especially in the upright position [31,49]; however, there is a lack of research on the bending posture [53]. Specifically, the focus has been more addressed on frontal plane analysis than on the sagittal plane [54]. Finally, SM and PG have recently been investigated in body perception in adolescents with idiopathic scoliosis [29].

### Limitations

The sample size, participants’ ages, morpho-structural characteristics, differences in spine misalignment, and level of physical activity represent limits in our work. Furthermore, Spinal Mouse^®^ and photogrammetry could be affected from measurement errors. Although the reliability and validity of both methods have been previously described, the occurrence of possible evaluation errors must be considered (markers positioning and angles investigation using photogrammetry or skin surface contact during Spinal Mouse^®^ analysis, for example). Future investigations are needed investigating the role of Spinal Mouse^®^ and photogrammetry during postural evaluation in adolescents with structural and non-structural hyperkyphosis and in relation to specific phases of treatment.

## 5. Conclusions

The present study suggests that photogrammetry can be considered an easy, inexpensive, and rapid tool for postural screening in adolescents with both structural and non-structural hyperkyphosis. Photogrammetry can be a useful alternative in the absence of other specific instruments, although it should be noted that photogrammetry is more suitable for screening rather than diagnosis. Photogrammetry is significantly correlated with the SM parameters when the analysis is taken during spine forward-bending. Specifically, the pelvic motion measured with photogrammetry can predict over 80% of the SM spine inclination. Differently, these instruments exhibited low correlations in standing positions. In conclusion, Kinesiologists and professionals involved in postural assessment are encouraged to use photogrammetry in bending as a first-line assessment tool to evaluate the sagittal spinal profile of adolescents.

## Figures and Tables

**Figure 1 jfmk-08-00068-f001:**
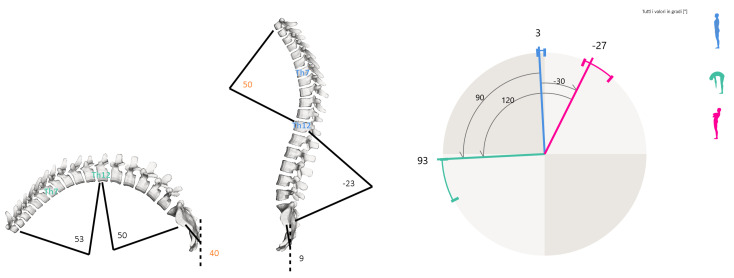
Example of Spinal Mouse report—thoracic and lumbar curvature (**left side**); body inclination (**right side**).

**Figure 2 jfmk-08-00068-f002:**
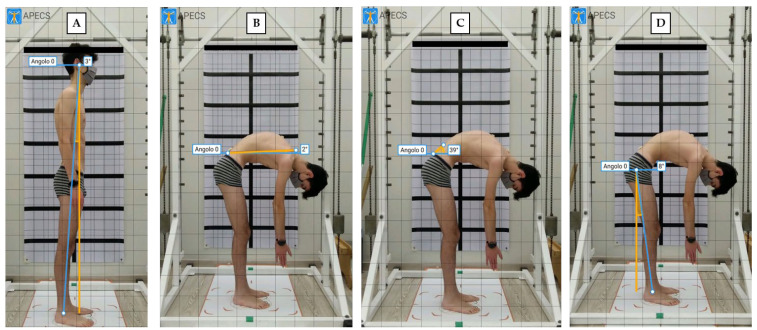
Photogrammetric analysis—(**A**) body vertical inclination (PG mall–tragus); (**B**) trunk flexion (PG trunk bend); (**C**) sacral inclination (PG sacrum bend); (**D**) hip position (PG hip bend).

**Figure 3 jfmk-08-00068-f003:**
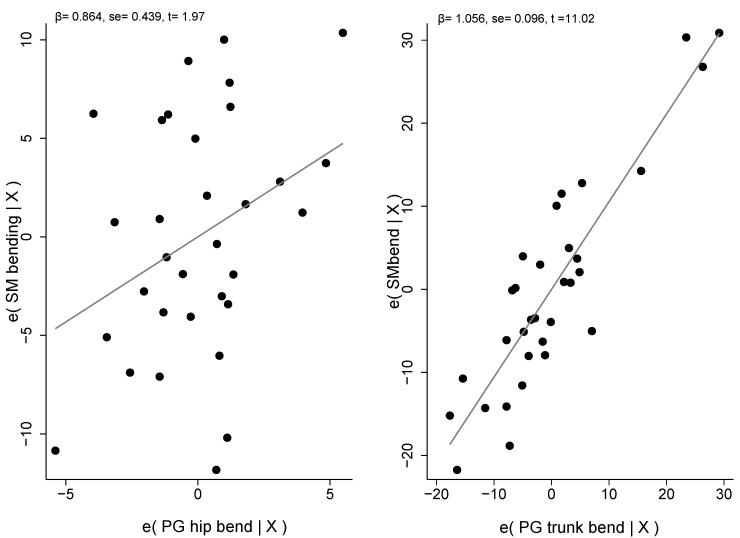
Errors’ scatterplots with SM and PG model regressors. Note: β, regressor coefficient; se, standard error; t, Student’s *t*-test.

**Figure 4 jfmk-08-00068-f004:**
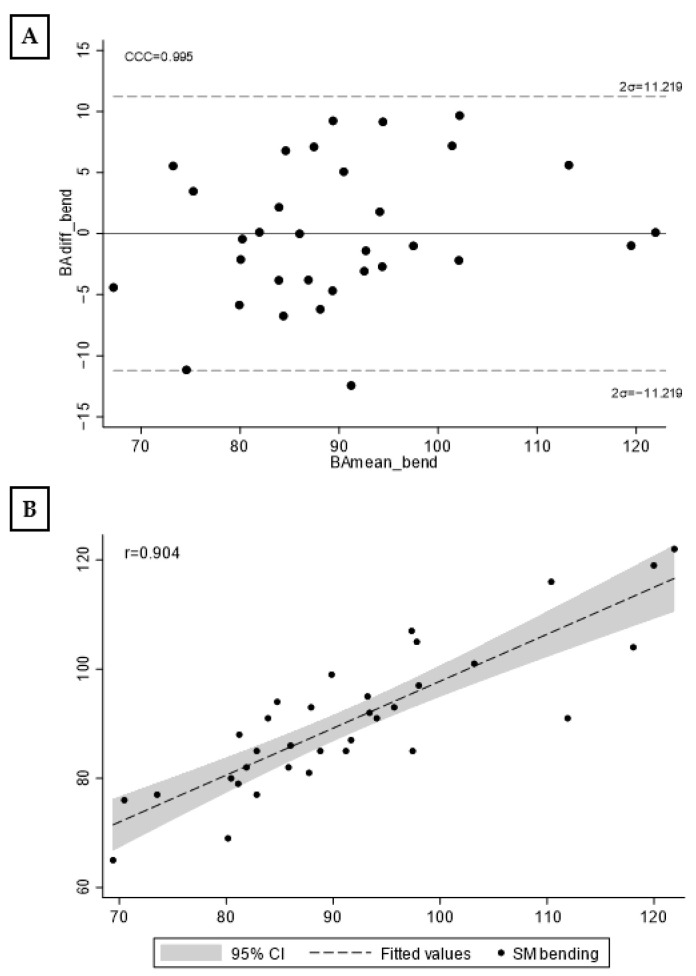
Bland–Altman plot (**A**) and scatterplot (**B**) with gold standard and new estimated model. Note: CCC, concordance correlation coefficient; σ, standard deviation; r, Pearson correlation coefficient; CI, confidence interval.

**Figure 5 jfmk-08-00068-f005:**
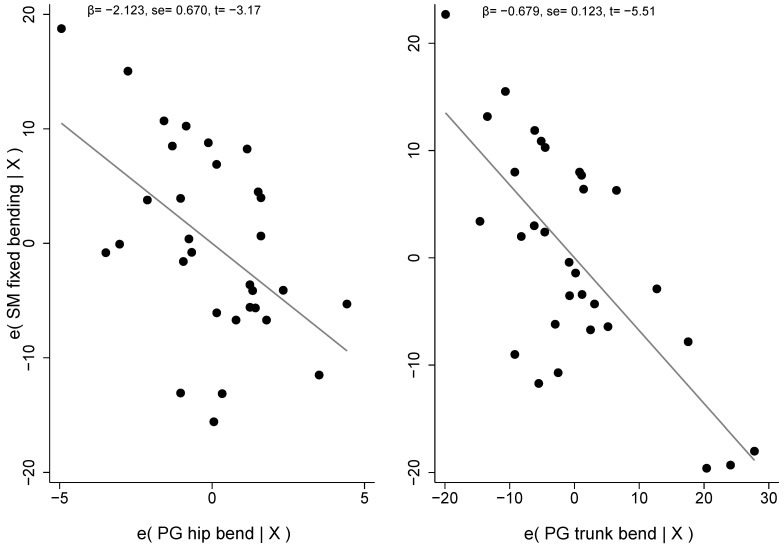
Errors’ scatterplots with SM fixed kyphosis and PG model regressors. Note: β, regressor coefficient; se, standard error; t, Student’s *t*-test.

**Figure 6 jfmk-08-00068-f006:**
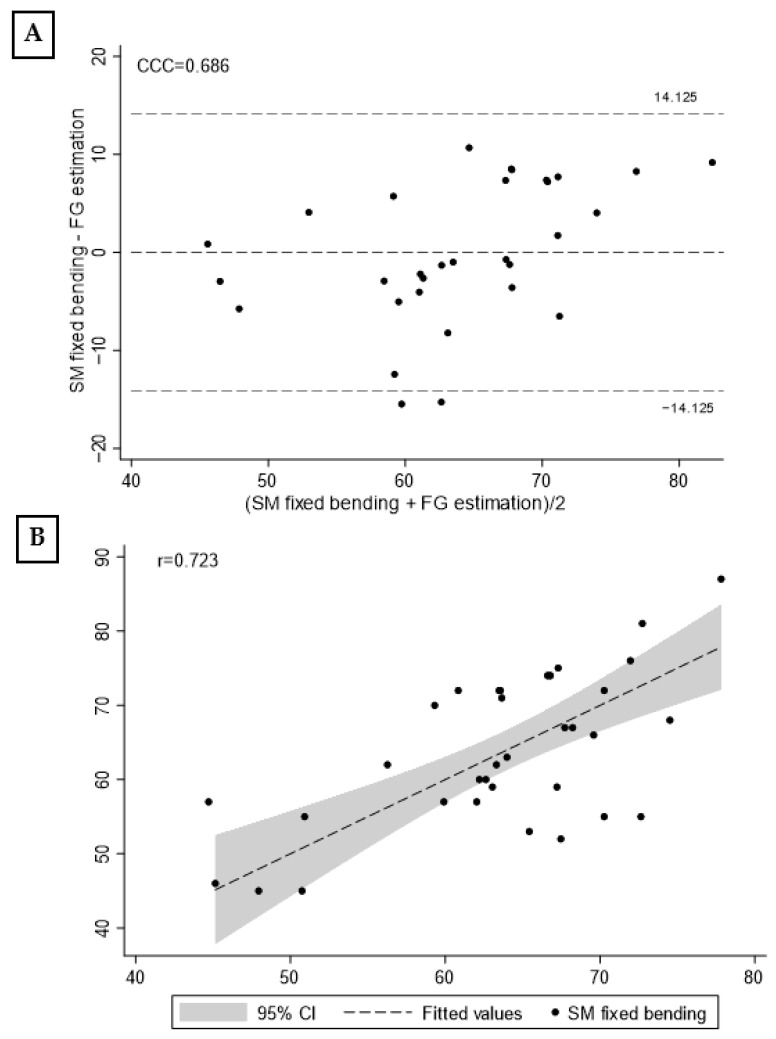
Bland–Altman plot (**A**) and scatterplot (**B**) with fixed Spinal Mouse in bending position and the new estimated model. Note: CCC, concordance correlation coefficient; σ, standard deviation; r, Pearson correlation coefficient; CI, confidence interval.

**Table 1 jfmk-08-00068-t001:** Descriptive statistics.

Variable (n = 34)	Mean	Std	Min	Max
Age [year]	13.06	1.84	10	18
Height [cm]	158.56	12.73	139	189
Weight [kg]	47.03	12.19	26	75
PG mall-tragus [°]	2.79	1.20	1	6
PG trunk bend [°]	91.71	13.35	68	121
PG sacrum bend [°]	37.15	12.02	4	55
PG hip bend [°]	8.71	2.69	3	15
SM stand [°]	1.68	2.81	−6	7
SM bend [°]	90.56	13.16	65	122
SM fixed stand [°]	45.94	8.24	29	62
SM phys. stand [°]	48.29	9.14	30	69
SM fixed bend [°]	63.71	10.23	45	87
SM phys. bend [°]	63.41	9.45	46	82
SM Rachid stand [mm]	457.18	44.02	394	593
SM Rachid bend [mm]	538.44	51.33	449	673

Note: n, number of observations; std, standard deviation; min, minimum value observed; max, maximum value observed; PG, photogrammetry; mall, malleolus; bend, bending; SM, Spinal Mouse; stand, standing; phys., physiological kyphosis.

**Table 2 jfmk-08-00068-t002:** Variables’ correlation matrix.

	PG Mall–Tragus	PG Trunk Bend	PG Sacrum Bend	PG Hip Bend	SM Stand	SM Bend	SM Fixed Stand	SM Phys. Stand	SM Fixed Bend	SM Phys. Bend	SM Rachid Stand	SM Rachid Bend
PG mall–tragus	-											
PG trunk bend	0.003	-										
PG sacrum bend	−0.076	−0.860 *	-									
PG hip bend	−0.176	−0.479 *	0.327	-								
SM stand	−0.045	−0.059	0.061	0.379 *	-							
SM bend	−0.041	0.839 *	−0.732 *	−0.237	0.119	-						
SM fixed stand	0.013	−0.204	0.330	−0.151	−0.098	−0.29	-					
SM phys. stand	0.036	−0.024	0.195	−0.182	−0.020	−0.17	0.908 *	-				
SM fixed bend	0.138	−0.526 *	0.559 *	−0.063	−0.076	−0.61 *	0.403 *	0.251	-			
SM phys. bend	0.060	−0.316	0.350 *	0.024	−0.149	−0.41 *	0.404 *	0.420 *	0.637 *	-		
SM Rachid stand	0.392 *	0.000	0.042	−0.222	−0.102	−0.08	0.206	0.319	0.121	0.272	-	
SM Rachid bend	0.273	−0.014	0.065	−0.243	−0.258	−0.1	0.180	0.243	0.228	0.332	0.914 *	-

Note: PG, photogrammetry; mall, malleolus; bend, bending; SM, Spinal Mouse; stand, standing; phys., physiological kyphosis; * *p*-value ≤ 0.05.

**Table 3 jfmk-08-00068-t003:** Best linear regression model for bending Spinal Mouse estimation.

Source	SS	df	MS	n	*F* _(2, 28)_	*p*	*R* ^2^	Adj. *R*^2^	Root MSE
Model	4513.86	2	2256.93	32	64.44	<0.001	0.816	0.804	5.918
Residual	1015.64	29	35.02						
Total	5529.5	31	178.37						
SM bending		*β*	SE	*t*	*p*	95% CI			
PG hip bend		0.86413	0.4394	1.97	0.059	−0.0346	1.763		
PG trunk bend		1.0557	0.0958	11.02	<0.001	0.86	1.252		
Intercept		−12.749	10.874	−1.17	0.251	−34.99	9.491		

Note: SS, squared sums; df, degrees of freedom; MS, squared means; n, number of observations; *F*, Snedecor–Fisher’s test; *p*, *p*-value; *R*^2^, the goodness-of-fit; Adj., adjusted; MSE, mean of squares error; *β*, regression coefficient; *t*, Student’s test; CI, confidence interval.

**Table 4 jfmk-08-00068-t004:** Best linear regression model for Spinal Mouse fixed kyphosis estimation.

Source	SS	df	MS	n	*F* _(2, 28)_	*p*	*R* ^2^	Adj. *R*^2^	Root MSE
Model	1701.8126	2	850.90628	31	15.29	<0.001	0.522	0.488	7.46
Residual	1558.0584	28	55.644943						
Total	3259.871	30	108.66237						
SM fixed bending		*β*	SE	*t*	*p*	[95% conf.	interval]		
PG hip bend		−2.122705	0.6704444	−3.17	0.004	−3.496048	−0.749361		
PG trunk bend		−0.678946	0.1232905	−5.51	<0.001	−0.931495	−0.426397		
Intercept		144.2784	15.00178	9.62	<0.001	113.5486	175.0081		

Note: SS, squared sums; df, degrees of freedom; MS, squared means; n, number of observations; *F*, Snedecor–Fisher’s test; *p*, *p*-value; *R*^2^, goodness-of-fit; Adj., adjusted; MSE, mean of squares error; *β*, regression coefficient; *t*, student’s test; CI, confidence interval.

## Data Availability

The data presented in this study are available on request from the corresponding author.

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
