# Peer review of "Relation between Photogrammetry and Spinal Mouse for Sagittal Imbalance Assessment in Adolescents with Thoracic Kyphosis"

_jfmk, 2023, doi:10.3390/jfmk8020068_

Round 1
Reviewer 1 Report
Overall well written. There are some typographical and grammatical errors, which need to be rectified
Abstract: The results need to be more clearly presented
Intro: Please include similar articles in the literature. Very long. The initial, general portion of intro can be reduced
Methods: elaborately explained
Please clearly explain in separate paragraphs what is photogrammetry and spine mouse
Results and Discussion:
Overall well presented. The discussion may be presented under subheadings for easier understanding
The study purpose is not clear. Please clearly explain
Pls compare the techniques with other modalities for scoliosis assessment. Why is this assessment methodology required?
Needs to be modified
Author Response
Overall well written. There are some typographical and grammatical errors, which need to be rectified
Authors: Dear reviewer, thank you for your comments. We appreciated them and improved our manuscript following your suggestions.
Abstract: The results need to be more clearly presented
Authors: We improved it in lines 20-24.
Intro: Please include similar articles in the literature. Very long. The initial, general portion of the intro can be reduced.
Authors: We improved the introduction. We reduced the initial and general portions of the introduction. Concerning the request for similar articles, we added in lines 88-91 the following sentence “Although radiographic and photogrammetric procedures have been previously investigated in order to quantify thoracic kyphosis and lumbar lordosis, the comparison between Spinal Mouse® evaluation and photogrammetry is lacking”.
Methods: elaborately explained
Please clearly explain in separate paragraphs what is photogrammetry and spine mouse.
Authors: The detailed descriptions of Spinal Mouse and Photogrammetry are already present in “2.2.1 Spine Analysis” and “2.2.2 Photogrammetric Postural Analysis”, respectively (lines 117 and 164).
Results and Discussion:
Overall, well presented. The discussion may be presented under subheadings for easier understanding.
Authors: Thanks for your suggestion. Despite considering your opinion, we prefer reporting discussions with no subheadings. However, we added one subheading for the limitations section.
The study purpose is not clear. Please clearly explain.
Authors: We added specific details to the purpose (lines 91-97).
Pls compare the techniques with other modalities for scoliosis assessment. Why is this assessment methodology required?
Authors: Dear Reviewer, the main topic of our investigation is the kyphotic posture. According to our perspective, it is difficult to write about comparisons with modalities for scoliosis assessment. If we could not understand your question, please explain it to allow us to meet your request.
Reviewer 2 Report
Dear Authors,
First of all, thank you for the opportunity to review your script. Certainly, a lot of work has gone into your study, yet I do not consider it fit for publication at this time.
My main criticism is that the innovation of the study is not clear. There are many studies on the use of photogrammetry for postural analysis, including kyphosis patients, for example:
Porto, A. B., & Okazaki, V. H. (2018). Thoracic kyphosis and lumbar lordosis assessment by radiography and photogrammetry: a review of normative values and reliability. Journal of manipulative and physiological therapeutics, 41(8), 712-723.
Porto, A. B., & Okazaki, V. H. A. (2017). Procedures of assessment on the quantification of thoracic kyphosis and lumbar lordosis by radiography and photogrammetry: A literature review. Journal of bodywork and movement therapies, 21(4), 986-994.
Similarly, there are many studies that have spinal mouse as their subject (you cite some of these as well). In this respect, it is not clear what is new about your study. This can also be seen in your conclusions. Some studies, which you also cite, have already drawn the same conclusions.
Secondly, I find it difficult to regard the Spinal Mouse as a kind of "gold standard" which you use to justify the usefulness of photogrammetry. For example, the Spinal Mouse has the crucial disadvantage that the skin has to be touched, which always reflexively leads to a postural change in some patients.
If you want to justify the benefit of photogrammetry by comparison with a "gold standard", then again only X-ray or 3D raster sterophotogrammetry would come into question. This, however, has already been investigated in other studies.
The following minor points also caught my attention:
Line 36: grammar?
Line 50: The publications of Dolphens et al. could be of interest here, e.g:
Posture class prediction of pre-peak height velocity subjects according to gross body segment orientations using linear discriminant analysis, M. Dolphens, B. Cagnie, P. Coorevits, A. Vleeming, T. Palmans and L. Danneels, European Spine Journal 2014 Vol. 23 Issue 3 Pages 530-535
Classification system of the normal variation in sagittal standing plane alignement, M. Dolphens, B. Cagnie, P. Coorevits, A. Vleeming and L. Danneels, Spine 2013 Vol. 38 Issue 16 Pages E1003-12
Line 63: „spine analysis of photogrammetry” à spine analysis by means of photogrammetry?
Line 67-73: The commercial systems mentioned are not the only ones on the market, nor are they the only reference systems. The measurement methods should be mentioned, but not exclusively two manufacturers.
Line 87: Reference is lacking
Table 1: Column “N” is unnecessary, mention N in the header.
Table 1: units are missing for all parameters
Fig 2 C: Angle is not well visible, possibly show detailed view
Results: Results are not clearly presented in the text, and results shown in tables and graphs are not addressed in the body text.
Author Response
Dear Authors,
First of all, thank you for the opportunity to review your script. Certainly, a lot of work has gone into your study, yet I do not consider it fit for publication at this time.
Authors: Dear reviewer, thank you for your comments. We appreciated them and improved our manuscript following your suggestions.
My main criticism is that the innovation of the study is not clear. There are many studies on the use of photogrammetry for postural analysis, including kyphosis patients, for example:
Porto, A. B., & Okazaki, V. H. (2018). Thoracic kyphosis and lumbar lordosis assessment by radiography and photogrammetry: a review of normative values and reliability. Journal of manipulative and physiological therapeutics, 41(8), 712-723.
Porto, A. B., & Okazaki, V. H. A. (2017). Procedures of assessment on the quantification of thoracic kyphosis and lumbar lordosis by radiography and photogrammetry: A literature review. Journal of bodywork and movement therapies, 21(4), 986-994.
Similarly, there are many studies that have spinal mouse as their subject (you cite some of these as well). In this respect, it is not clear what is new about your study. This can also be seen in your conclusions. Some studies, which you also cite, have already drawn the same conclusions.
Authors: Dear reviewer, the reported studies investigated only the spinal curvature (thoracic, lumbar, pelvic tilt) in specific sports players or correlated anthropometric parameters with kyphosis and lordosis degrees in the school population. However, we improved the introduction and added the above-mentioned references in line 90. The focus of our study is on the comparison between Spinal Mouse and photogrammetry concerning specific variables of the sagittal plane.
Similar conclusions have been reported in the discussion section in relation to spinal degrees only (kyphosis and lordosis), not about Spinal Mouse – Photogrammetry comparison.
Secondly, I find it difficult to regard the Spinal Mouse as a kind of "gold standard" which you use to justify the usefulness of photogrammetry. For example, the Spinal Mouse has the crucial disadvantage that the skin has to be touched, which always reflexively leads to a postural change in some patients.
Authors: Since the Spinal Mouse has been reported as a reliable and valid tool for this population, we considered it a practice and useful device to compare with photogrammetry (practical and less expensive). The introduction and discussion described the gold standard for sagittal assessment (radiographic methods) and defined the limits of both measurements.
If you want to justify the benefit of photogrammetry by comparison with a "gold standard", then again only X-ray or 3D raster sterophotogrammetry would come into question. This, however, has already been investigated in other studies.
Authors: Compared to gold standard measurements (X-Ray), Spinal Mouse can be used in several body postures. For this reason, a comparison with photogrammetry in the same positions (upright standing and forward trunk bending, in present research) is possible and could be interesting (from our point of view). Consequently, conclusions are limited to this relation (photogrammetry has been widely investigated in an upright position).
The following minor points also caught my attention:
Line 36: grammar?
Authors: We cancelled that line.
Line 50: The publications of Dolphens et al. could be of interest here, e.g:
Posture class prediction of pre-peak height velocity subjects according to gross body segment orientations using linear discriminant analysis, M. Dolphens, B. Cagnie, P. Coorevits, A. Vleeming, T. Palmans and L. Danneels, European Spine Journal 2014 Vol. 23 Issue 3 Pages 530-535
Classification system of the normal variation in sagittal standing plane alignement, M. Dolphens, B. Cagnie, P. Coorevits, A. Vleeming and L. Danneels, Spine 2013 Vol. 38 Issue 16 Pages E1003-12
Authors: Thank you for your suggestion. We added the first study you mentioned in line 44.
Line 63: „spine analysis of photogrammetry” à spine analysis by means of photogrammetry?
Authors: Thank you, we corrected it.
Line 67-73: The commercial systems mentioned are not the only ones on the market, nor are they the only reference systems. The measurement methods should be mentioned, but not exclusively by two manufacturers.
Authors: We corrected it in lines 70-73.
Line 87: Reference is lacking.
Authors: we added it.
Table 1: Column “N” is unnecessary, mention N in the header.
Table 1: units are missing for all parameters.
Fig 2 C: Angle is not well visible, possibly show detailed view.
Authors: Thank you for your comment. We improved Table 1 and Figure 2 according to your suggestions.
Results: Results are not clearly presented in the text, and results shown in tables and graphs are not addressed in the body text.
Authors: Thanks for your comment. Results have been improved in lines (233-238; 242-254; 257-260; 284-288).
Reviewer 3 Report
This is very interesting study for physiotherapists and other clinical professionals. My main concer about this study is that the photogrammetry should be compared to very reliable method like RTG (if occepted by Ethical Committee) or 3D motion analisys like Vicon or Qualisys. This should be considered in the future studies and in the manuscript in the study limitations.
Following minor revisions are needed:
- line 58 - please write some examples of these factors;
- line 73 - the reference seems to be lost (...RGB camera and it seems to be reliable as well (REF).);
line 89 - what do you mean under "It requires basic or advanced skills from the user"? In which systems only basic skills are required and in which the advanced ones?;
- table 2 - correlations of the same variables (r=1) should be rather marked as "-" - it makes no sense to write "1"; it also hard to read numbers that are in two lines;
- line 349 - "Krawky and colleagues" should be "Krawky, et al."
Author Response
This is very interesting study for physiotherapists and other clinical professionals. My main concer about this study is that the photogrammetry should be compared to very reliable method like RTG (if occepted by Ethical Committee) or 3D motion analisys like Vicon or Qualisys. This should be considered in the future studies and in the manuscript in the study limitations.
Authors: Dear reviewer, thank you for your comments. We appreciated them and improved our manuscript following your suggestions.
Following minor revisions are needed:
line 58 - please write some examples of these factors.
Authors: We deleted this line.
line 73 - the reference seems to be lost (...RGB camera and it seems to be reliable as well (REF)).
Authors: Thank you, we added the reference.
line 89 - what do you mean under "It requires basic or advanced skills from the user"? In which systems only basic skills are required and in which the advanced ones?
Authors: Our sentence was referred to the “software friendly user experience” that can be subjective. Anyway, we modified the sentence. Line 93.
table 2 - correlations of the same variables (r=1) should be rather marked as "-" - it makes no sense to write "1"; it also hard to read numbers that are in two lines.
Authors: Thank you for your comment. We improved Table 2 according to your suggestions. Although we want to follow your advice, we did not understand what you meant by “numbers that are in two lines”.
line 349 - "Krawky and colleagues" should be "Krawky, et al."
Authors: Corrected at line 360.
Round 2
Reviewer 1 Report
The recommended changes have been made. The manuscript can be accepted in the present form
Overall, the manuscript reads well
Author Response
Dear reviewer, thank you for supporting our manuscript.
Reviewer 2 Report
Dear Authors,
thank you very much for the revision of your manuscript. You have implemented most of the suggestions. There are still some points that should be improved:
Figure 1: please mention the reference for this figure (or was it self-generated?)
Table 1: only two digits after the decimal point
Line 228 ff: “Similar measures were found between fixed and physiological Spinal Mouse evaluation, ….”
I don't understand the meaning of this sentence - what values are we talking about?
Line 194 f: To be able to better understand the angles measured by photogrammetry and to be able to reproduce the measurements, please indicate the positioning of the anatomical marker points / marker balls.
Fig 2 C: the measured angle is not shown very clearly - where are the marker points positioned? With the spatially close position of the two marker points, the measurement error is large. Please refer to this in the discussion and indicate the magnitude of the measurement error if possible.
Line 332 f: the results of the cited study do not logically fit this passage or your study.
Line 358 ff: The differences between the two measuring systems for the parameter 'Sagittal vertical inclination' are interesting and should be discussed in more detail. Since inclination according to Dolphens et al. can be related to possible pain symptoms, it seems to be a particularly important parameter. Here, photogrammetry is certainly superior to SM because it measures in an absolute coordinate system, whereas SM can only measure using an internal coordinate system and measurement errors add up. You should present this interesting point in more detail, because it again makes the advantages of your measuring method clearer.
Line 388: You should include the possible disadvantages of measuring with the SM (compared to photogrammetry) due to touching the skin surface (with possible reflex postural change) in the limitations.
Line 400 ff: You would also have to mention for which parameters the correlation was low, i.e. there is no comparability of the measurement systems!
Author Response
Rev: Dear Authors,
Thank you very much for the revision of your manuscript. You have implemented most of the suggestions. There are still some points that should be improved:
Figure 1: please mention the reference for this figure (or was it self-generated?)
A.: Dear reviewer, figure 1 was extracted from IDIAG M360 software using data from one participant of this study (as reported in lines 157-159).
Table 1: only two digits after the decimal point
A.: we corrected it as you requested.
Line 228 ff: “Similar measures were found between fixed and physiological Spinal Mouse evaluation, ….”
I don't understand the meaning of this sentence - what values are we talking about?
A.: The values are referred to 2 different Spinal Mouse measures of thoracic spine curvature (kyphosis angle). Both measures are described in line 152 and 153. In particular, Fixed Spinal defines the thoracic curvature between first (T1) and last thoracic vertebra (T12) in standing or bending position (Fixed Stand and Fixed Bend, respectively), while Physiological Spinal describes the curvature in relation to upper and lower kyphosis limits located by software (in standing and bending position).
Line 194 f: To be able to better understand the angles measured by photogrammetry and to be able to reproduce the measurements, please indicate the positioning of the anatomical marker points / marker balls.
A.: Dear reviewer, to avoid too much information in the text, we referred to previous studies that reported the same method we have used. As regards anatomical marker points, you can find further details in the study of Stolinski et al. [43]. We hope you can share our point of view.
Fig 2 C: the measured angle is not shown very clearly - where are the marker points positioned? With the spatially close position of the two marker points, the measurement error is large. Please refer to this in the discussion and indicate the magnitude of the measurement error if possible.
A.: As reported in the above comment, the marker points are well-debated in the studies we cited, as well as the measurement errors. Figure 2 is just an example to easier interpret how photogrammetry evaluations are usually assessed in bending and standing positions. Marker points were identified and evidenced as described in lines 194-196.
Line 332 f: the results of the cited study do not logically fit this passage or your study.
A. The study of Muyor et al. has been reported because bending test was executed and analyzed using Spinal Mouse in similar way. For this reason, the results have been briefly exposed to better understand their work. Furthermore, Muyor reported that pelvic tilt and lumbar mobility are significantly related to bending test score. Consequently, hip motion and sacral inclination were investigated using photogrammetry.
Line 358 ff: The differences between the two measuring systems for the parameter 'Sagittal vertical inclination' are interesting and should be discussed in more detail. Since inclination according to Dolphens et al. can be related to possible pain symptoms, it seems to be a particularly important parameter. Here, photogrammetry is certainly superior to SM because it measures in an absolute coordinate system, whereas SM can only measure using an internal coordinate system and measurement errors add up. You should present this interesting point in more detail, because it again makes the advantages of your measuring method clearer.
A. This is an interesting point of discussion. Since our sample didn’t report thoracic or lumbar pain, we supposed that sagittal inclination could be more influenced by musculoskeletal profile. We improved the limits of both devices in a specific paragraph (see below)
Line 388: You should include the possible disadvantages of measuring with the SM (compared to photogrammetry) due to touching the skin surface (with possible reflex postural change) in the limitations.
A.: We added the limits of Spinal Mouse and Photogrammetry in paragraph 4.1, as suggested
Line 400 ff: You would also have to mention for which parameters the correlation was low, i.e. there is no comparability of the measurement systems!
A.: We added a phrase in line 202, as suggested.
Round 3
Reviewer 2 Report
The authors have implemented most of the reviewers' suggestions. The discussion could have been expanded somewhat regarding the advantages and disadvantages of both correlated measurement methods.